# Industrial technology progress, digital finance development and corporate risk-taking: Evidence from China's listed firms

Xiya Wu[1], Yanghui Liu[1], Biru Xia[2]*

1 School of Economics and Management, East China Jiaotong University, Nanchang, Jiangxi, China,
2 School of Business Administration, Jiangxi University of Finance and Economics, Jiangxi, China

* 18707997578@163.com

## Abstract

Industrial technological progress, as an essential industrial-technological and institutional phenomenon, brings with it the possibility of high profits for firms but also implies new norms and rules of competition, which affect the willingness and propensity of firms to bear the costs of undertaking venture capital projects. This study empirically investigates the causal impact of industrial-technological progress on corporate risk-taking and the mechanism of digital financial growth on the relationship between the two, based on data from China's A-share listed businesses from 2011 to 2020. This paper finds that (1) industrial technological progress improves enterprise risk-taking levels. Moreover, digital financial development has an incentive effect on industrial technological progress and enterprise risk-taking levels. (2) Industrial technological progress under digital financial development generates financing constraint relaxation effects, input capital return enhancement effects, and innovation performance incentive effects, increasing enterprise risk-taking. (3) The positive moderating effect of digital financial development on the relationship between industrial technological progress and the risk-taking level of enterprises in the eastern regions and enterprises in the high-tech industry is more prominent. The study's findings provide a theoretical foundation and policy insights on the crucial elements of industrial-technological progress and enterprises' increased ability to take risks throughout the development of digital finance.

## 1. Introduction

Industrial technology progress is the main driving force and core element to accelerate industrial upgrading. It is well known that technological progress arises from a combination of other technologies [1]; that is, technological progress is essentially constantly being constructed, aggregated, and integrated from previously existing technologies to form new technologies. Industrial technological progress, on the other hand, comes from the combination of other technologies at the industrial level, when the boundaries of technology space and the shares of different types of technologies in technology space are optimized and combined among industries, and the efficiency of factor utilization at the industrial level is increased.,

**Data Availability Statement:** All relevant data are within the paper and its Supporting Information files.

**Funding:** This work was funded by Jiangxi National Social Science Fund (Project: 22YJ50D) and Jiangxi University Humanities and Social Science

Fund (Project: JJ22212), both awarded to YL. The funders had no role in study design, data collection and analysis, decision to publish, or preparation of the manuscript.

**Competing interests:** The authors have declared that no competing interests exist

forming industrial technological progress. Industrial technological progress, as an essential socio-technical and institutional phenomenon, forms the operational basis, rules, and field of technological application of micro-enterprises, forms the institutional environment of enterprises, and determines the behavior of enterprises in choosing risky investments in technological innovation and technological application,i.e., whether they are willing to bear the corresponding risks. The selection of venture capital initiatives by businesses in their quest for high profits, which reflects their readiness and propensity to absorb the costs of taking on additional risk, is one way risk-taking is expressed [2]. This is also the key for companies to enhance their core competencies and succeed. The improvement of their risk-taking level can help companies increase R&D investment, improve ROI, and enhance enterprise value, which is a critical factor in improving social productivity and promoting economic growth [3–5].

Industrial technology progress is the process of socializing technology applications due to market supply and demand and cost-benefit selection mechanisms, which provides enterprises with the opportunity to increase their core competitiveness and chase high profits and induces them to make risky investments in technology research and development and technology application around the industry. However, the uncertainty of technology innovation, technology application investment, and future commercialization levels may also bring increased risk, and the level of enterprise risk-taking becomes one of the critical considerations for enterprise decision-makers to make enterprise-risky investments. Does technological progress in the industry affect the level of risk-taking by firms? Related studies have yet to be conducted.

Industrial technology advancement and corporate venture capital, among others, require capital allocation efficiency beyond the traditional financial model, and digital finance development has become a capital allocation model that meets such efficiency and information requirements. Digital finance has strengthened the traditional financial model's weaknesses, broadened the financing channels of technology research and development enterprises, made it possible for enterprises that used to have difficulty obtaining loans from banks to obtain financial funds [6], and effectively increased the motivation and success rate of enterprise innovation. With the help of digital technologies like big data, the internet of things, artificial intelligence, and cloud computing, digital finance can better address the issue of high operating costs and risk premiums brought on by information asymmetry [7, 8], maximize the effectiveness of resource allocation [9], and present new opportunities for business growth. Corporate development cannot be achieved without financial support, and how to use digital technology to improve corporate performance has been the focus of theoretical attention [10]. To this end, does the development of digital finance bring industrial technology advances to enhance corporate risk-taking further and effectively? What is the mechanism? Rationalizing the mechanism of action is conducive to correctly understanding and analyzing the relationship between industrial technological progress and enterprise risk-taking under the current digital finance model.

The primary contributions of this paper are as follows: First, it is the first study of the microeconomic consequences of industrial-technological progress. Different from the existing literature, which mainly assesses the economic consequences of technological progress from the perspectives of industry, employment, and the population labor force, only a small amount of literature deals with the relationship between technological progress and economic growth. As an essential issue at the industrial level, industrial-technological progress reflects the effects and forms of technological progress at the industrial level, etc. However, research on industrial technological progress has yet to be carried out, and its micro-economic consequences are even rarer. At present, academics generally focus on industrial upgrading and macro-economic consequences. However, the issue of industrial-technological progress, which is intrinsically and logically related to industrial upgrading, has yet to attract due attention, and

research on the impact on micro-enterprises has not been involved. This paper is the first to study the mechanism of the impact of industrial-technological progress on enterprise risk-taking, bridging the gap between theoretical and empirical research on the economic effects of technological progress on micro-enterprises. Second, this paper is a pioneering exploration of the mechanism by which industrial technological progress affects enterprise risk-taking under the digital finance model. Previous research has dealt with the relationship between the digital financial model and enterprise risk. However, research on the interaction between industrial technological advancement and firm risk-taking capacity is scarce. This paper is the first to incorporate the digital finance model, a significant influence in reality, into the research framework of industrial-technological progress on corporate risk-taking and to investigate its mechanism of action in depth. In comparison, most of the existing literature analyzes corporate risk-taking from the perspectives of ownership structure, management traits, and corporate governance mechanisms. Lastly, this paper further extends the macro-influencing factors in the traditional theoretical model of corporate risk-taking, exploring and enriching the key points to provide a theoretical basis and policy inspiration for improving corporate risk-taking capacity in the process of industrial technology progress and digital finance development.

## 2. Theoretical foundation and hypotheses development

### 2.1 Corporate risk-taking and industrial technology progress

Industrial technology is the form of technology evolution at the industrial level, and some scholars have fitted the evolutionary process of industrial technology with cluster succession dynamics and found that there is a nonlinear evolutionary process of industrial technology from a discrete distribution of non-interference to a highly clustered network structure of synergistic evolution [11]. Industrial technology plays a fundamental role in promoting industrial upgrading, while industrial upgrading also drives industrial technology progress, which is an inevitable phenomenon in the process of industrial upgrading. There is a complex feedback mechanism between the two, and industrial technology progress and industrial upgrading go hand in hand.

Industrial technological progress has complexity, and firms that adapt to industrial technological progress have a higher level of risk-taking. Technological progress follows the process of technological innovation, selection, and diffusion, which includes the expansion of the boundaries of the technological space and changing the composition of the share of different types of technologies in the established technological space [12]. The process of technological progress is accompanied by complexity features such as nonlinearity, interaction, trajectory, emergence, and system-in-system [13–15], and industrial technological progress likewise presents complexity at the industrial level, constituting an external institutional environment for firms, and firms that adapt to industrial-technological progress have a higher level of risk-taking. Specifically, first, industrial-technological progress can optimize resource allocation among enterprises and boost enterprise risk-taking. Industrial technological progress results from the synergistic evolution of industrial systems and technological progress. When enterprises adapt to industrial-technological progress, their production and operation efficiency and quality can be improved, and the competitiveness of enterprises continues to be generated and improved spontaneously. In a certain period and under certain resource constraints, the enhancement of competitiveness generation and improvement ability determines the direction of resource flow, and the flow of resources to enterprises adapting to industrial-technological progress improves the risk-taking level of these enterprises. Second, technological progress in industry can promote the diffusion of technological knowledge and increase the degree of risk-taking by businesses. The industrial chain links many enterprises in different links

together and makes the development of enterprises not only depend on their own technology level but also be influenced by the overall technology level of the chain, and the industrial-technological progress brings about the diffusion of technological knowledge among enterprises, and the quality and speed of enterprise technology application depends on the level of industrial-technological progress, which reduces the cost and risk of enterprises independent R&D and stimulates enterprises to release more resources for risky activities such as research and development [16], the overall technological progress of the whole industry promotes technology integration and application among enterprises, enhances the technological cooperation and risk-sharing mechanism of enterprises in the overall industry chain, and strengthens the risk-taking capacity of enterprises in the industry chain. Lastly, industrial-technological progress brings more benefits to enterprises and raises enterprise risk-taking. Industrial technology progress increases the level of technologicalization of enterprises, and enterprise innovation investment increases accordingly, which also means the emergence of new products and models, intensifying market competition. Enterprises that cannot adapt to the application of new technologies as early as possible often lose in market competition, while enterprises that can adapt to new technologies obtain higher performance and return on invested capital and the degree of business risk-taking increases.

Industrial technology is divisible and systematic, and enterprises applying new industrial technology can often share the risk with other enterprises in the industry chain and improve risk-taking. Improving the level of science and technology has led to the continuous refinement and deepening of the division of labor, which has led to the decomposition of elements, processes, and procedures. Even a particular link in the industry can be divided into several tiny links, and a large number of intermediate products have emerged, which has dramatically increased the technical divisibility of the industry and played a role in promoting vertical separation. The economic activities engaged in by enterprises are only a particular stage of the division of labor at the industrial level, and there is a natural connection between the various links of industrial technology, showing the interdependence of industrial technology and organization and synergistic evolution characteristics. Any intermediate links and processes of industrial technology are an end after the role of social selection mechanisms, often breaking through the traditional enterprise boundaries and forming a complete technological system through market contracts. Through technical cooperation with other enterprises, enterprises in the industry can fill the industrial technology gap and share the risk, improving enterprise risk-taking.

Industrial technologies are market-selective, and enterprises tend to obtain higher returns and increase their level of risk-taking by applying new industrial technologies. Industrial technology is a process of socialization of technology application, which is mainly realized through the market, and the overriding principle of market selection is whether technology is economical and applicable,i.e., cost-benefit evaluation constitutes the primary criterion for selection. Without cost-benefit judgment, technology cannot enter the industrial level. Industrial technologies that enter the industrial level through the market elimination mechanism often bring high profits and controllable risks to enterprises, and the willingness and ability of enterprises to invest in the corresponding risks are enhanced, raising the level of enterprise risk-taking. Based on the above analysis, this paper proposes the hypotheses H1:

*Industrial technological progress significantly affects the level of corporate risk-taking.*

## 2.2 Industrial technology progress, digital finance development, and corporate risk-taking

Industrial technological progress and enterprise risk-taking levels are highly dependent on the external digital financial environment. Digital finance can be seen as a financial form and environment corresponding to the digital economy arising from the superposition of finance and technology. Serving the real economy is the foundation of the digital finance concept, with financial technology acting as the means. Digital finance development can encompass various aspects such as digital banking, digital payments, digital supply chain finance, digital inclusive finance, digital insurance, and so on. The foundation on which digital financial development rests is big data and credit information. Digital finance development involves all kinds of information subjects, R&D institutions, and data interoperability between financial institutions and various information subjects, breaking down information barriers. Financial institutions can use big data, cloud computing, and other advanced information technology to more accurately assess the efficiency of technology research and development and enterprise venture capital funding and realize information sharing. Financial institutions can use big data, cloud computing, and other advanced information technology to more accurately assess the efficiency of technology research and development and enterprise venture capital funding, realize information sharing, and alleviate the information asymmetry between fund suppliers, R&D institutions, and corporate venture capital. Fund suppliers have a comprehensive understanding of corporate information to facilitate the precise implementation of policies, improve the efficiency of approval of corporate venture capital projects, help industrial technology progress, and improve corporate risk-taking. With the development of digital technology, digital finance has produced a new normal and a new industry, and digital financial products have become more prosperous and more complex, which can more effectively meet the business needs of enterprises.

The rise of digital finance can satisfy businesses' desire to take on more risk. Corporate risk choice is influenced by the institutional environment [17–19], the traditional financial model, which generates agency problems such as information asymmetry and credit discrimination [20, 21], and financing constraints [22], where firms lack sufficient funds nor the willingness to take risks, reducing the level of corporate risk-taking. The development of digital finance makes it relatively easy to monitor and use funds; the cost of obtaining information on factors such as funds is reduced; enterprises also have easy access to external funds, and their willingness to invest is enhanced. More funds and willingness to take risks correspond to higher risk-taking levels, and at this time, enterprises are less likely to give up high-risk and high-return investment opportunities, and they tend to be willing to pay more to chase high profits in the market. On the one hand, the development of digital finance makes financial institutions and other capital suppliers rely on digital technology-driven and massive data mining to obtain more information on enterprise risk investment, be able to more accurately assess enterprise investment projects and financial status, effectively solve the problem of information asymmetry that exists in the acquisition of funds by enterprises, alleviate the problem of information asymmetry, and form a more stable source of funds that are expected to reduce the uncertainty of future funds. Entrepreneurs' confidence in venture investment increases, and their mood tends to be positive. Business decision-makers have more security and self-confidence, and changes in their psychological characteristics can further affect their choice of investment risk [23, 24]; On the other hand, digital finance development can lessen the dependence of traditional finance on physical outlets and reduce the cost of capital for enterprises [25]; especially digital money can increase the coverage of financial services and lower service costs [26], which is conducive to the optimal allocation of financial assets. The issue of capital mismatch

under the conventional financial model has a major improvement effect, which can enhance the effectiveness of resource allocation across the board and ease the pressure of corporate financing restrictions. The expansion of corporate capital availability and corporate capital ownership also influences the decision to pursue risky investment projects. Digital financial development affects firms' willingness to take risks, and firms tend to chase risky investment projects with high profits. Consequently, the following hypotheses in this research are:

**H2:** *Digital finance development has a moderating effect on the relationship between industrial technological progress and corporate risk-taking.*

**2.2.1 Corporate innovation performance.** The knowledge formed by enterprise innovation activities will attract competitors to imitate and learn in the market and gradually become public knowledge with externalities. The R&D cost of competitors is almost zero, while the R&D enterprises have a high cost but cannot get due compensation. The digital finance model facilitates government subsidies to play an essential form of knowledge protection and compensation in enterprise innovation activities, and the development of digital finance improves the efficiency of enterprise capital turnover and reduces enterprise capital costs accordingly. Particularly in the case of collaborative R&D in industrial technology, the development of digital finance recognizes and supports the knowledge of R&D companies and reduces the uncertainty and risk of R&D capital investment [27]. More importantly, digital financial development also has a positive role in the promotion and application of industrial technology, through which it facilitates technological transactions and spillovers and reduces the cost of evolving private knowledge into public knowledge. Digital financial development, to a certain extent, has the function of private knowledge protection and compensation for R&D costs, incentivizing innovation activities, and improving innovation performance. In addition, the development of digital finance has led to the rationalization of factor pricing in the market and the flow of innovative resources to enterprises with innovative advantages, which effectively reduces the cost of R&D for enterprises and promotes their innovative inputs and substantive innovative outputs. Moreover, out of the consideration of reducing R&D costs and optimizing the structure of investment, enterprises are willing to invest their R&D resources in their innovative activities and obtain more innovative performance, thus forming a virtuous circle. In summary, this paper proposes the following research hypothesis:

**H2a:** *Under the role of digital financial development, industrial-technological progress generates incentives for firms' innovative performance and improves the level of firms' risk-taking.*

**2.2.2 Corporate return on capital.** The development of digital finance has changed the way in which enterprise investment and financing information is produced, transmitted, and exchanged. The direct interface between mass social capital and enterprises can reduce the degree of information asymmetry, reduce the irrational behavior of enterprises, increase the probability of deviation between actual returns and investment expectations, and increase the return on capital enterprises invest. The direct interface between mass social capital and enterprises can reduce the degree of information asymmetry, reduce the irrational behavior of enterprises, increase the probability of deviation between actual returns and investment expectations, and increase the return on capital enterprises invest. According to information economics, differences in the amount of information possessed will evolve into an imbalance in the distribution of benefits. Investors who lack sufficient information will engage in irrational investment behavior, thus accelerating the transmission and diffusion of risk, which is not conducive to increased corporate risk-taking. Under the digital finance model, enterprises can

adapt to advancing industrial technology more quickly with the help of digital information, have a more comprehensive understanding of relevant technology investment, R&D investment, technology mergers and acquisitions, and other venture capital projects, improve the rationalization of scientific decision-making, reduce irrational investment behavior due to information asymmetry, increase the probability of deviation between actual returns and investment expectations, and improve the return on investment capital. In addition, enterprises in the digital finance model tend to have relatively abundant resource support, and the management of enterprises accordingly becomes less risk-averse and tends to favor riskier but highly profitable projects. Notably, the potential market competitive advantage that new technologies may bring is more favored, and management tends to grasp better the investment opportunities brought by technological advancements. As a result, their willingness to take risks in investment activities rises, and as the return on invested capital also improves, this level of risk-taking by enterprises also rises. Accordingly, it can be hypothesized that:

**H2b**: *Under the role of digital financial development, industrial-technological progress generates the effect of corporate return on capital growth and improves corporate risk-taking.*

**2.2.3 Relaxation of financing constraints.**   In order to support new technological applications, adapt to new business models, and adopt new modes as industrial technology advances, businesses require a sizable and steady cash flow. Additionally, they may have to take short-term financial risks as well as long-term market risks. If enterprises face more significant financing constraints at this time, they often reduce high-risk and high-cost innovation expenditures and instead invest funds in projects with little investment but quick effect, thus leading to or resulting in a lower level of new technology applications. Under the traditional financial model, information asymmetry leads to external financing constraints [28], reducing the level of risk-taking and forcing companies to scale back their investment in new technology systems, while under the digital financial model, financial institutions and other funding providers combine digital technology with financial innovation, not only reducing information asymmetry to achieve efficient funding supervision and borrowing decision basis for funding providers but also providing efficient and convenient financial services for funding demanders and reducing the cost of funding. The digital finance model achieves the goal of providing financial services to enterprises in the real economy and easing financing constraints, especially in terms of stimulating industrial technological progress. Specifically, industrial-technological progress directs the flow of resources to enterprises that actively adapt to the new technological system and alleviates resource mismatches between enterprises.

On the one hand, the growth of digital finance has increased the range of financial services available to the real economy, increasing the likelihood that more businesses will be approved for credit and other forms of financial assistance. Traditional credit institutions usually base their lending on enterprises' collateralizable assets. Enterprises lacking collateral often face more significant financing constraints. At the same time, the development of digital finance enables the assessment of the long-term development prospects of such enterprises. It can benefit more enterprises, especially SMEs, which have difficulties in obtaining traditional credit. On the other hand, the development of digital finance has decreased the information asymmetry between the supply and demand of funds, and financial institutions have increased the level of corporate risk-taking by using technologies like the internet, artificial intelligence, and cloud computing to screen and evaluate businesses while more effectively supplying funds. Therefore, this study can be conducted by considering the following hypotheses:

*H2c*: *Under the role of digital financial development, industrial-technological progress generates the relaxation effect of corporate financing constraints and improves the corporate risk-taking level.*

# 3. Model construction and variable selection

## 3.1 Model construction

**3.1.1 Benchmark model.** To examine how industrial technological progress affects business risk-taking, we employ the model shown below:

$$Riski_{i,t} = \alpha_0 + \alpha_1 \sum Tfp_{i,t} + \alpha_2 DFD_{i,t} + \sum Control + \sum Industry + \sum Year + s, t \quad (1)$$

Where the dependent variable $Risk_{i,t}$ denotes the firm's risk-taking level, the independent variable $Tfp_{i,t}$ denotes the level of technological progress of the industry, *and Control* represents the control variable. In addition, the model also controls the industry fixed effect *Industry*, and the annual fixed effect *Year.$\varepsilon_{i,t}$* is the error term.

Control variables were taken from previous studies on factors affecting firm risk-taking. At the firm level, traditional factors influencing corporate risk-taking include profitability ($Roa_t$), firm size (*$Size_t$*), firm growth ($Growth_t$), top ten shareholders shareholding ($Ten_t$), dual employment ($Dual_t$), cash ratio ($Cashratio_t$), and management expense ratio ($Cost_t$). At the same time, macro control variables such as the regional gross domestic product ($Gdp_t$) and the local consumer pricing index ($Cpi_t$) are added to consider the influence of macro factors on corporate risk-taking. The specific variable definitions in the text are shown in Table 1.

**3.1.2 Moderation effect.** To test hypothesis 2, this paper uses a moderating effect model to verify the incentive effect of digital financial development on industrial technological progress and enterprise risk-taking. Model (1) introduces the interaction term between industrial

**Table 1. Definition of variables.**

| Variable | Statistic | Description |
|---|---|---|
| Corporate risk-taking | *Risk* | The volatility of the firm's surplus(ROA)during the observation period |
| Industrial technological progress | *Tfp* | Total factor productivity by industry |
| Digital finance development | *DFD* | Provincial-level digital financial inclusion indexes are logarithmically processed |
| Firm performance | *Roa* | The rate of return on average corporate assets. |
| Corporate growth | *Growth* | (Operating income for the quarter-Operating Revenue for the previous quarter)/Operating Revenue for the previous quarter |
| Corporate size | *Size* | Natural logarithm of the total assets of the firms. |
| Ownership concentration | *Ten* | Major top 10 shareholders to total shares |
| CEO duality | *Dual* | Equals one if the CEO and Chairperson are the same person and zero otherwise |
| Cash holding ratio | *Cashratio* | Closing balance of cash and cash equivalents/total assets |
| Cost of management expenses | *Cost* | The ratio of corporate management expenses to primary business income |
| The regional consumer price index | *Cpi* | The consumer price index of the province where the company |
| Gross regional product | *Gdp* | The logarithm of the gross regional product of the province in which the company |

technological progress and digital financial development (*Tfp_DFD*) and constructs model (2):

$$Risk_{i,t} = \beta_0 + \beta_1 Tfp_{i,t} + \beta_1 DFD_{i,t} + \beta_3 Tfp\_DFD_{i,t} + \sum Control + \sum Industry$$
$$+ \sum Year + s, t \tag{2}$$

## 3.2 Measurement of variables

**3.2.1 Corporate risk-taking.** This paper uses the volatility of corporate earnings as an indicator of risk-taking. Commonly used indicators in the established literature to measure firm risk-taking include surplus volatility [29, 30], stock volatility [3, 31], cash flow volatility [21], firm R&D expenses [19, 31], etc. Corporate cash flows are more strongly impacted by seasonal factors because of the high synchronization and volatility of the Chinese stock market. Corporate accounting requirements impact corporate R&D expenses more since non-innovative enterprises have fewer values available. Drawing on José María et al.'s (2019) approach, this paper uses the volatility of corporate surplus to measure corporate risk-taking [32]. The calculation follows: ROA equals the company's net income divided by total assets at year-end. This study uses a rolling three-year observation period (T = 3, period t to t+2)to compute the standard deviation of the firm's industry-year adjusted return on total assets.

**3.2.2 Industrial technology progress.** Previous studies have mainly used total factor productivity to measure technological progress. Total factor productivity measurement methods are mainly divided into non-parametric and parametric methods. The DEA-Malmquist index methods, which are more traditional but cannot test the applicability of the frontier surface and do not account for the influence of random factors on the measurement results, dominate the non-parametric approach. The parametric approach, however, can get around the drawbacks above. The Solow residual approach, the latent variable method, and the stochastic frontier analysis (SFA) method are three subcategories of parametric methods. SFA considers, to some extent, the impact of the random error term on total factor productivity [33]. Drawing on Luo and Li's (2023) study, this paper adopts the total factor productivity of industry measured by the SFA parametric method to measure the technological progress of industry [34]. The output indicator is real GDP; the base period is 1990. Moreover, the input indicators are the number of employees in society and fixed assets accounted for by the perpetual inventory method with a parameterized depreciation rate of 9.6%.

**3.2.3 Digital financial development.** Digital financial inclusion has developed into the most representative business model of digital finance. In light of this, we use the digital finance index created by the Institute of Digital Finance at Peking University in this paper, referencing the methodology of Guo et al. (2020), to reflect the development of digital finance in China and select provincial data with logarithmic processing [35].

**3.2.4 Control variables.** We select the control variables from three aspects of firm characteristics, financial position, and corporate governance while also adding macro-level influences. The specific control variables are selected as follows: (1) Firm performance (2) Corporate growth (3) Corporation size (4) Ownership concentration (5) CEO duality (6) Cash holding ratio (7) Cost of management expenses (8) The regional consumer price index (9) The gross regional product.

## 3.3 Sample selection and data

The research sample for this paper covers A-share-listed Chinese firms from 2011 to 2020. To screen the sample by earlier research, we use several criteria, including the exclusion of listed financial sector companies, businesses in the ST or ST* category, samples with outliers in the

data, and samples with missing variables. Our final sample after these stages includes 26200 industry-year observations. The China Statistical Yearbook and the China Stock Market and Accounting Research (CSMAR) database are used to get all data. To counteract the impact of extreme values, we additionally lower the tails of the 1% and 99% quartiles for all continuous variables in this study.

## 4. Results

### 4.1. Descriptive statistics

As shown in Table 2, the mean and standard deviation of enterprise risk-taking (*Risk*) are 0.010 and 0.010. The minimum value of industrial-technological progress (*Tfp*) is 5.778, and the maximum value is 8.160, indicating that the level of industrial-technological progress varies greatly. The larger the value of total factor productivity, the higher the level of industrial-technological progress possessed by the enterprise, which helps the enterprise to utilize the enterprise capital better, thus obtaining a better enterprise risk-taking ability. The minimum value of digital financial development (*DFD*) is 3.013, and the maximum value is 6.068, which indicates that the degree of development of the digital financial model varies significantly in different provinces. It is possible to study the comparison of the digital financial model of the sample enterprises in different industries and its impact on industrial technological progress and enterprise risk-taking.

### 4.2 Analysis of Benchmark results

Ordinary least squares (OLS) are utilized in this study to regress the data. In Table 3, Column (1) displays the regression analysis results for model (1), which indicate that the industrial-technological progress coefficient is positive and significant at the 1% level. Column (2) displays the regression findings of the model (2), which indicate that the interaction term between industrial technical advancement and digital financial development has a positive coefficient and is significant at the 1% level. Adding relevant control variables to the original, columns (3) and (4) are added, and their significance stays unchanged. This means there is a significant positive correlation between industrial technological progress and enterprise risk-taking, that hypothesis 1 is accurate, and that the digital financial models have a positive moderating effect. The higher the level of industrial-technological progress, the stronger the ability of enterprises to take risks. Hypothesis 2 is proven.

**Table 2. Descriptive statistics.**

| Variable | Obs | Mean | Std. dev. | Min | Max |
|---|---|---|---|---|---|
| Risk | 26200 | 0.010 | 0.010 | 0.001 | 0.058 |
| Tfp | 26200 | 6.646 | 0.486 | 5.778 | 8.160 |
| DFD | 26200 | 5.255 | 0.741 | 3.013 | 6.068 |
| Roa | 26200 | 0.040 | 0.062 | -0.261 | 0.206 |
| Growth | 26200 | 0.021 | 0.777 | -0.119 | 92.909 |
| Size | 26200 | 22.126 | 1.321 | 19.666 | 26.173 |
| Ten | 26200 | 0.592 | 0.158 | 0.013 | 1.012 |
| Dual | 26200 | 0.725 | 0.447 | 0.000 | 1.000 |
| Cashratio | 26200 | 0.173 | 0.140 | 0.011 | 0.679 |
| Cost | 26200 | 0.093 | 0.076 | 0.009 | 0.483 |
| Cpi | 26200 | 102.518 | 1.024 | 101.032 | 105.636 |
| Gdp | 26200 | 10.395 | 0.716 | 7.873 | 11.615 |

**Table 3. Baseline regression analysis.**

|  | (1) | (2) | (3) | (4) |
|---|---|---|---|---|
|  | *Risk* | *Risk* | *Risk* | *Risk* |
| *Tfp_DFD* |  | 0.0007*** |  | 0.0005*** |
|  |  | (4.12) |  | (2.95) |
| *Tfp* | 0.0004** | 0.0005*** | 0.0021*** | 0.0022*** |
|  | (2.27) | (2.63) | (11.40) | (11.61) |
| *DFD* | 0.0007** | 0.0008** | 0.0002 | 0.0002 |
|  | (2.21) | (2.52) | (0.60) | (0.83) |
| *Roa* |  |  | 0.0076*** | 0.0076*** |
|  |  |  | (5.58) | (5.57) |
| *Growth* |  |  | 0.0002 | 0.0002 |
|  |  |  | (1.60) | (1.61) |
| *Size* |  |  | -0.0010*** | -0.0010*** |
|  |  |  | (-19.74) | (-19.82) |
| *Ten* |  |  | 0.0029*** | 0.0029*** |
|  |  |  | (7.15) | (7.13) |
| *Dual* |  |  | -0.0004*** | -0.0004*** |
|  |  |  | (-3.03) | (-3.04) |
| *Cashratio* |  |  | 0.0071*** | 0.0071*** |
|  |  |  | (11.17) | (11.05) |
| *Cost* |  |  | 0.0057*** | 0.0056*** |
|  |  |  | (5.48) | (5.41) |
| *Cpi* |  |  | 0.0004*** | 0.0004*** |
|  |  |  | (2.83) | (2.86) |
| *Gdp* |  |  | 0.0001 | 0.0001 |
|  |  |  | (0.82) | (0.84) |
| *_cons* | 0.0124*** | 0.0115*** | -0.0224 | -0.0234 |
|  | (7.15) | (6.67) | (-1.45) | (-1.52) |
| Industry | Yes | Yes | Yes | Yes |
| Year | Yes | Yes | Yes | Yes |
| Adj. R2 | 0.0712 | 0.0718 | 0.1162 | 0.1164 |
| N | 26200 | 26200 | 26200 | 26200 |

*$p < 0.05$

**$p < 0.01$

***$p < 0.001$

## 4.3 Heterogeneity analysis

This paper examines the impact of industrial-technological progress on firm risk-taking, considering differences in policy support, guidance, and protection across regions and industries. It can generate endogeneity issues. Based on Han and Li's (2013) split-sample heterogeneity validation method, the sample is divided into regional and industry levels based on regional and industry characteristics [36]. The underlying assumption of the split-sample heterogeneity validation method is that it is difficult for firms to change their regional location and industry characteristics. Thus, regional location and industry attributes can be regarded as exogenous, and the impact of industrial-technological advancement on firms' risk-taking varies according to regions and industries, which can essentially rule out the endogeneity problem arising from

the government's selective introduction of supportive, guiding, and protective policies for firms in regions and industries.

**4.3.1 Regional differences.** Based on the National Bureau of Statistics (NBS) criteria in China, we divided the eastern and central-western regions into two groups. The results, shown in Columns (1) and (3) of Table 4, show that industrial-technological progress has a strong effect on enterprises in the eastern region but a less specific effect on those in the central and western regions. The digital financial model has a positive moderating effect on risk-taking for enterprises in the eastern regions, as shown by columns (2) and (4). The eastern area firms' coefficient of the interaction term is significantly positive at the 1% level, indicating that the development of digital finance has a more significant moderating effect on the enterprises in the eastern regions than those in the central and western regions.

**4.3.2 Industry differences.** The high-tech businesses examined in this paper include those manufacturing pharmaceuticals, instruments, computers, communication and other electronic equipment, aerospace, railroad, ship, and other transportation equipment, among others. Non-high-tech enterprises are businesses in other sectors. The coefficients of the interaction terms for high-tech industries are at the 1% level of statistical significance, as shown in columns (6) and (8) of Table 4, suggesting that the digital finance model has a more positive moderating effect on the relationship between industrial technological advancement and the capacity for risk-taking of high-tech firms. Since there is a significant difference between high-tech and non-high-tech enterprises, the coefficient on the interaction term for non-high-tech firms is insignificant, suggesting that the positive moderating effect of digital financial development on high-tech industries is more significant.

## 4.4 Robustness tests

**4.4.1 Replacement of core variables.** This study switched to using the range (max-min) of the industry's annual adjusted return on total assets to calculate risk-taking. Furthermore, it re-runs the regression analysis on industrial technological progress, digital financial development, and corporate risk-taking. The test results in Columns (1)-(2) of Table 5 are consistent with the previous results in Table 3.

**4.4.2 Add the possible control variables.** Although our study controls as much as possible for other factors affecting firm risk-taking, there may still be endogeneity problems due to the omission of certain control variables. Considering that corporate governance issues such as the structure of the company's board of directors and compensation management may affect the company's level of risk-taking, this paper adds the total compensation of the top three executives (*Compen*), the size of the board of directors (*Board*), and the compensation payable to the employees (*Salary*) as the control variables in the model (1) and then conducts regression analyses. The regression results in Columns (3)-(4) of Table 5 are compatible with our prior results, which were given in Table 3, which increases the dependability of our research conclusions.

**4.4.3 Instrumental variable regression.** Analyzed at the theoretical level, industrial-technological progress, as a phenomenon in the external system of enterprises, affects their risk investment decisions,i.e., risk-taking. As a micro-variable, industrial-technological progress is difficult to influence macro-variables, so it is difficult to establish a reverse causality. Drawing on Huang and Li's (2006) method, each province and region's foreign market proximity variable is used as an instrumental variable in the endogeneity test [37]. Foreign market proximity reflects market environment differences, and the position of the front, middle, and back ends of the technological division of labor in the industrial chain and different technological positions will depend on market environment differences. Therefore, foreign market proximity

**Table 4. Heterogeneity analysis.**

|  | Eastern Region | | Central and Western region | | High-tech industries | | Other industries | |
|---|---|---|---|---|---|---|---|---|
|  | **(1)** | **(2)** | **(3)** | **(4)** | **(5)** | **(6)** | **(7)** | **(8)** |
|  | *Risk* | *Risk* | *Risk* | *Risk* | *Risk* | *Risk* | *Risk* | *Risk* |
| Tfp_DFD |  | 0.0005** |  | 0.0004 |  | 0.0013*** |  | -0.0003 |
|  |  | (2.29) |  | (1.22) |  | (4.24) |  | (-1.07) |
| Tfp | 0.0019*** | -0.0008 | 0.0018*** | -0.0002 | 0.0020*** | -0.0049*** | 0.0016*** | 0.0029** |
|  | (6.89) | (-0.64) | (6.30) | (-0.14) | (7.79) | (-2.97) | (5.23) | (2.30) |
| DFD | 0.0001 | -0.0033** | 0.0014 | -0.0011 | 0.0001 | -0.0086*** | 0.0010* | 0.0027 |
|  | (0.18) | (-2.04) | (1.52) | (-0.55) | (0.17) | (-4.03) | (1.96) | (1.54) |
| _cons | 0.0067 | 0.0236 | -0.0564* | -0.0434 | -0.0277 | 0.0161 | -0.0054 | -0.0141 |
|  | (0.29) | (0.98) | (-1.71) | (-1.30) | (-1.07) | (0.59) | (-0.23) | (-0.58) |
| Control | Yes | Yes | Yes | Yes | Yes | Yes | Yes | Yes |
| Industry | Yes | Yes | Yes | Yes | Yes | Yes | Yes | Yes |
| Year | Yes | Yes | Yes | Yes | Yes | Yes | Yes | Yes |
| Adj. R2 | 0.1042 | 0.1044 | 0.0967 | 0.0967 | 0.1000 | 0.1009 | 0.1005 | 0.1005 |
| N | 18111 | 18111 | 8089 | 8089 | 15144 | 15144 | 11056 | 11056 |

*$p < 0.05$

**$p < 0.01$

***$p < 0.001$

constitutes a crucial external environment for industrial technological progress, which can screen the formation of local industrial technology and industrial technological progress with high correlation but does not directly affect the level of enterprise risk-taking and is not correlated with the level of enterprise risk-taking and the residual term of the model so that it can be used as an instrumental variable for industrial technological progress. At the same time,

**Table 5. Robustness tests.**

|  | **(1)** | **(2)** | **(3)** | **(4)** | **(5)** | **(6)** | **(7)** | **(8)** |
|---|---|---|---|---|---|---|---|---|
|  | *Risk* | *Risk* | *Risk* | *Risk* | *Risk* | *Risk* | *Risk* | *Risk* |
| Tfp_DFD |  | 0.0005*** |  | 0.0009*** |  | 0.0025** |  | 0.0007*** |
|  |  | (2.79) |  | (3.08) |  | (2.27) |  | (4.01) |
| Tfp | 0.0021*** | 0.0022*** | 0.0040*** | 0.0040*** | 0.0184** | 0.0179** | 0.0021*** | 0.0022*** |
|  | (11.38) | (11.57) | (11.47) | (11.69) | (2.36) | (2.33) | (10.92) | (11.28) |
| DFD | 0.0000 | 0.0001 | 0.0003 | 0.0004 | 0.0020 | 0.0022 | 0.0001 | 0.0002 |
|  | (0.07) | (0.28) | (0.56) | (0.80) | (1.12) | (1.23) | (0.29) | (0.58) |
| _cons | -0.0221 | -0.0231 | -0.0370 | -0.0390 | -0.4243*** | -0.4254*** | -0.0266 | -0.0277* |
|  | (-1.42) | (-1.49) | (-1.30) | (-1.37) | (-4.29) | (-4.30) | (-1.63) | (-1.70) |
| Control | Yes | Yes | Yes | Yes | Yes | Yes | Yes | Yes |
| Industry | Yes | Yes | Yes | Yes | Yes | Yes | Yes | Yes |
| Year | Yes | Yes | Yes | Yes | Yes | Yes | Yes | Yes |
| Adj. R2 | 0.1161 | 0.1164 | 0.1197 | 0.1200 | 0.2100 | 0.2100 | 0.1216 | 0.1221 |
| N | 25969 | 25969 | 26200 | 26200 | 19,526 | 19,526 | 22913 | 22913 |

*$p < 0.05$

**$p < 0.01$

*** $p < 0.001$

industrial-technological progress has typical path-dependent characteristics, and the level of industrial-technological progress in the previous period will affect industrial-technological progress and risk investment decisions in the next period. Drawing on Yuan and Xie (2014), the explanatory variables are lagged by one period of data, i.e., the indicator of industrial-technological progress lagged by one period is used as an instrumental variable [38]. After using the two-stage least squares method to carry out the above regression, the regression results are shown in Column (5)-(6) in Table 5, and the results are in line with the previous section, indicating that the study's findings are valid.

**4.4.4 Selection of subsamples.** Considering the impact of the COVID-19 pandemic, in this paper, the regression test was re-run by excluding the sample during the epidemic, and the results in Columns (7)-(8) were consistent with the previous study.

## 5. Mechanism analysis

The previous theoretical analysis shows that improvement in industrial technological progress under the influence of digital financial development results in innovation performance incentive effects, the growth effect of return on invested capital, financing constraints relaxation effects, and an increase in enterprise risk-taking. This paper builds on Nunn.'s (2007) and Wang et al.'s (2010) work and builds a model based on model (2) by adding the interaction terms of mechanism variables with industrial technological progress and digital financial development to carry out mechanism testing [39, 40]:

$$Risk_{i,t} = \delta_0 + \delta_1 Meantfp\_DFD\_Mech_{i,t} + \delta_2 Meantfp - DFD_{i,t} + \sum Control + \sum Industry + \sum Year + s_{,t} \quad (3)$$

Where $Mech_t$ is the firm's mechanism variable, drawing on most practices in academia, this paper is based on measuring innovation performance in terms of the number of patents invented by firms plus one to take the logarithm ($Patent_t$); the variable $ROIC_t$ is the return on invested capital; and the SA index measures the degree of relaxation of financing constraints ($SA_t$). The greater the SA value, the greater the degree of financing constraints.

### 5.1 Digital financial development and industrial technological progress generate incentives for firms' innovative performance and increase corporate risk-taking

Based on the test outcomes for model (3), as displayed in columns (1) in Table 6, the coefficients of the interaction terms of industrial-technological progress, digital financial development, and innovation performance are significantly positive at the 1% level. Therefore, hypothesis 2a is true, which demonstrates that innovation performance is a mechanism for improving the enterprise's capacity for risk under the control of a digital financial model.

### 5.2 Digital financial development and industrial technological progress generate an increase in the return on corporate capital and increase the level of corporate risk-taking

The model (3) test results, as presented in Column (2) of Table 6, indicate that the coefficients of the interaction terms of digital financial development, industrial-technological progress, and return on invested capital are significantly positive at the 5% level, indicating the validity of Hypothesis 2b, which suggests that the increase in return on invested capital is the mechanism through which the action of industrial-technological progress modifies the development of digital financial development, thereby enhancing the risk-taking capacity of enterprises.

**Table 6. Mechanism analysis.**

| | (1) | (2) | (3) |
|---|---|---|---|
| | *Risk* | *Risk* | *Risk* |
| Tfp_DFD_Patent | 0.0029*** | | |
| | (2.62) | | |
| Tfp_DFD | 0.0004*** | | |
| | (11.19) | | |
| Tfp_DFD__ROIC | | 0.0077** | |
| | | (2.30) | |
| Tfp_DFD | | 0.0004*** | |
| | | (2.65) | |
| Tfp_DFD__SA | | | -0.0029** |
| | | | (-2.50) |
| Tfp_DFD | | | 0.0005*** |
| | | | (7.66) |
| _cons | -0.0100 | -0.0234 | -0.0082 |
| | (-0.60) | (-1.52) | (-0.52) |
| Control | Yes | Yes | Yes |
| Industry | Yes | Yes | Yes |
| Year | Yes | Yes | Yes |
| Adj. R2 | 0.0991 | 0.1157 | 0.1194 |
| N | 25995 | 26149 | 24880 |

$^{*}$p $<$ 0.05

$^{**}$p $<$ 0.01

$^{***}$p $<$ 0.001

### 5.3 Digital financial development and industrial technological progress generate a relaxation of corporate financing constraints and increase the level of corporate risk-taking

The test results for model (3), displayed in columns (3) of Table 6, show that the interaction term's coefficients are significantly negative at the 5% level for industrial technological advancement, digital financial development, and financing constraints index. Hypothesis 2c is confirmed, showing that the rise in the financing constraint relaxation degree is a mechanism of how industrial technical advancements contribute to improving an organization's capacity for taking risks under the control of digital financial development.

## 6. Conclusions and recommendations

The study results show that technological progress will significantly increase enterprise risk-taking throughout the process of industrial upgrading. At the same time, the digital financial model will have a favorable moderating effect on industrial technological progress and enterprise risk-taking. The mediation effect test with moderating effect shows that the improvement of enterprise innovation performance and return on invested capital, as well as the relaxation of financing constraints under the influence of the digital financial model, is the primary way industrial technological progress affects enterprise risk-taking. Split-sample mechanism tests reveal that the digital finance model has a higher moderating impact on Eastern area businesses and businesses in high-tech industries than on Central and Western region businesses and businesses in non-high-tech sectors.

Scientific and technological self-reliance and self-improvement must be strengthened against China's current bottleneck in crucial core technologies. This paper is based on the close relationship between industrial technological progress and enterprise risk-taking, the digital financial model as an entry point, the logical framework of industrial-technological progress on corporate risk-taking is constructed through three channels, namely, relaxation of financing constraints, corporate innovation performance, and return on invested capital. It constructs a logical framework of industrial-technological progress in enterprise risk-taking, explores the black box between industrial-technological progress and enterprise risk-taking, and provides insights for promoting enterprises to play the role of innovation's main body and helping China's industrial technology to be self-reliant and self-improving. The results of this study support the digital financial model's beneficial moderating influence on the association between industrial technological progress and company risk-taking. Additionally, this study adds to the body of knowledge on micro-firm innovation from the standpoint of macro-industrial technology. It offers suggestions on how to improve firm risk-taking through the development of industrial technology. The following implications flow from the study's findings:

1. From the enterprise level. First, enterprises should take a more positive, open, and inclusive attitude towards developing new industrial technology forms and utilize the advantages of industrial technology resources and information to enhance enterprise risk-taking further and cultivate their core competitiveness. Enterprises should decide whether to carry out the combination of industry and finance and how to carry out the combination of industry and finance according to their advantages and the regional and industrial institutional environment, and when the mode of production-financing combination causes the enterprise's operation risk to be more significant, it should consider the positive exit method. Secondly, it is necessary to strengthen the level of internal corporate governance, optimize the power configuration within the group, and actively explore and innovate new modes of industry-finance integration to prevent and control its risks to alleviate the agency problems caused by industry-finance integration. Enterprises should actively communicate with financial institutions and assign experienced managers to interface with them to form a closed loop of information communication. At the same time, allocating financial assets should align with the enterprise's solvency to minimize the financial risks brought by the combination of production and financing.

2. At the level of digital financial models. First, in the process of promoting the digital financial model, we should not only broaden and deepen the use of digital finance, but we should also improve the digital financial facilities so that the various aspects of digital finance can work in concert to increase the risk-taking of businesses. Concentrating on the balanced growth of digital finance among businesses in various regions and industries while actively promoting the sector is crucial. Second, financial institutions should fully utilize digital financial technology to provide complete financial support to businesses with significant financing needs and outstanding innovation potential and increase the effectiveness of allocating credit resources. Financial institutions should fully use high-end big data technologies to create a credit assessment system integrating data integration, risk evaluation, and intelligent matching. A robust institutional framework is the only assurance that digital finance will have a creative incentive effect. Lastly, because it is a relatively new financial model, digital finance lacks the requisite institutional and legislative oversight, which could result in systemic financial problems and not be conducive to protecting the innovative achievements of digital creative enterprises. On the other hand, a scientific prudential supervision mechanism should be established, and technical means should be utilized to

monitor the risks in the financial market and protect the legitimate rights and interests of enterprises in the process of financing innovative projects.

3. As far as government departments are concerned, they should not only guide and promote industrial technology development but also prevent and resolve the risks of industrial integration by formulating relevant regulations and adopting appropriate guidance for enterprises with higher business risks. The quality of service offered by local governments should be improved, and they should ensure that the services they offer to various businesses are innovative. Government agencies should be flexible in their use of tax breaks, financial subsidies, and other preferential policies to allay business concerns, assist some businesses in undergoing digital transformation, and direct other businesses to actively undergo digital transformation in order to achieve industrial digitization by serving as a model case study to assist businesses in resolving the enduring issue of insufficient risk-taking. At the same time, relevant businesses must also actively safeguard the interests of investors, improve the quality of information disclosure, and minimize related-party transactions and other irregularities in trading practices. Additionally, reforms of the capital and bond markets should be firmly pushed to create a highly efficient capital market system. This will expand the options for financing businesses and boost the effectiveness of capital allocation. The role of the government as an intermediary in sustaining and enhancing technological innovation performance must be expanded. When formulating industrial technology-related policies, the government must classify them, accurately implement them, and purposefully reduce the risks associated with technological investment that different types of enterprises must deal with. Moreover, it is targeted to play a positive moderating role of digital finance in the impact of technological progress on enterprise risk-taking in the East, to explore ways to resolve the risks of technological application faced by enterprises in the central and western regions, as well as to promote the leading role of high-technology industries in the improvement of technological innovation performance, to satisfy the diversified needs of the enterprise groups.

## Supporting information

**S1 Data.**
(XLSX)

## Author Contributions

**Conceptualization:** Yanghui Liu, Biru Xia.

**Data curation:** Xiya Wu.

**Formal analysis:** Xiya Wu.

**Methodology:** Xiya Wu.

**Writing – original draft:** Xiya Wu.

**Writing – review & editing:** Yanghui Liu, Biru Xia.

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
