## [Decision Letter · Decision Letter 0]

2 Jan 2024

PONE-D-23-41757Industrial Technology Progress ,Digital Finance DevelopmentAnd Corporate Risk-Taking: Evidence from China's listed firmsPLOS ONE

Dear Dr. Wu,

Thank you for submitting your manuscript to PLOS ONE. After careful consideration, we feel that it has merit but does not fully meet PLOS ONE’s publication criteria as it currently stands. Therefore, we invite you to submit a revised version of the manuscript that addresses the points raised during the review process.

We look forward to receiving your revised manuscript.

Kind regards,

Han Lin

Academic Editor

PLOS ONE

Journal Requirements:

Additional Editor Comments:

The theoretical contributions could be better articulated, linking them back to the identified conversants and the research methods employed. This, in tandem with an explicit depiction of how the results impact our understanding of the research topic, would substantially elevate the study's contribution. It would be beneficial if the authors could identify the papers they aim to contribute to and delineate their novel addition to the existing discourse.

It might be helpful with new citations of up-to-date and high-quality papers on the subjects covered in manuscript and engaged the previous literature in this journal.

Also, the author(s) should carefully format the manuscript following the guidance of the journal, such as the headers, references, and so on. There are mistakes in the citation (reference), such as missing volume, issue and page.

The authors are recommended to examine and proof the language. In addition, there are some typos and grammatical errors that need further attention.

Reviewers' comments:

Reviewer's Responses to Questions

**Comments to the Author**

1. Is the manuscript technically sound, and do the data support the conclusions?

Reviewer #1: Yes

Reviewer #2: Yes

2. Has the statistical analysis been performed appropriately and rigorously? 

Reviewer #1: No

Reviewer #2: Yes

3. Have the authors made all data underlying the findings in their manuscript fully available?

Reviewer #1: Yes

Reviewer #2: Yes

4. Is the manuscript presented in an intelligible fashion and written in standard English?

Reviewer #1: Yes

Reviewer #2: Yes

5. Review Comments to the Author

Reviewer #1: The relationship between industrial technological progress and corporate risk-taking is important and interesting question. This paper makes a profound study through empirical research, as well as the mechanism of the role of digital financial development on the relationship between them. Through model and data analysis, it is concluded that industrial technological progress not only improves firms' ability to raise the level of risk-taking, but also generates three incentive effects through the incentive role of digital financial development, further expanding its impact on corporate risk-taking. According to regional and industry differences, the development of digital finance has a more pronounced positive moderating effect on industrial technological progress and risk-taking level among enterprises in the central and western regions as well as enterprises in high-tech industries. The main highlight of the article is that the impact of industrial technological progress on the behavior of microenterprises, this type of problem is extremely important in both academia and practice, but there is little literature on this field, while adding the hot topic of digital finance, the article is attractive and has the potential to be published, but suffers for several serious limits:

1. There is no literature reference for measuring industrial technological progress by total factor productivity, so it is suggested to supplement.

2. As a matter of routine, the organization of the paper should be provided at the end of introduction.

3. Benchmarking model, moderating effect model, etc. only have equations without more detailed explanations of variables.

4. The research sample is fine, but there is no literature background in section 3.1.

5. The comprehensibility of the tables in the paper is low, mainly because of the following problems: (1) none of the tables indicate the dependent variable; (2) the tables are disclosed without textual explanations.

5. It is suggested that the endogeneity testing should be further supplemented.

6. Although the case of China is presented, there is no explanation as to whether this is a global problem for many countries, which warrants a study of the Chinese case as its justification.

7. Are there any limitations to the thesis? What about future research perspectives?

8. Some statements in the essay need to be improved and standardize the terminology.

Reviewer #2: Dear author,

Based on the data of China's A-share listed companies from 2011 to 2020, the paper empirically investigates the impact of industrial technological progress on firms' risk-taking and the role played by digital financial development in it. The study finds that industrial technological progress promotes the increase in the level of corporate risk-taking. Digital financial development has a positive moderating effect on the relationship between industrial technological progress and the level of corporate risk-taking. Under the role of digital financial development, industrial technological progress has produced financing constraint alleviation effect, input capital return enhancement effect and innovation performance incentive effect. In addition, industrial technological progress has a significant impact on the level of corporate risk-taking in the eastern region and among non-high-tech firms, while digital financial development has a more pronounced positive moderating effect on the relationship between industrial technological progress and the level of risk-taking among firms in the central and western regions as well as among firms in high-tech industries. Throughout the existing research, the literature on the impact of industrial technological progress on firms' micro-behavior is extremely rare; therefore, the paper enriches the research literature on the microeconomic consequences of industrial technological progress and the factors influencing firms' risk-taking, and it also provides marginal evidence for the research literature on digital finance.

In conclusion, the theoretical study of this article on industrial technology is a very novel and interesting topic with research value and some innovative significance. But the article has some worthwhile improvements, as follows:

1. The split-sample test does not provide an analysis relevant to the topic, but only a split-sample regression of the data, and the rationale for the split-sample needs to be further elaborated.

2. The article needs to make some additional explanations on the model construction.

3. The robustness test is too simple, it is recommended to add more tests to ensure the reliability of the conclusion.

4. The purpose is not clearly stated in the abstract, and the methodology is not described.

5. The analysis of the regression results is rather simple and does not analyze the deeper reasons behind the results.

6. the period 2011-2020 is the beginning of the covid pandemic, will the data be affected by this?

7. The spacing between words is inconsistent in some parts of the article, and the formatting needs to be improved.

Sincerely.

6. PLOS authors have the option to publish the peer review history of their article (what does this mean?). If published, this will include your full peer review and any attached files.

Reviewer #1: **Yes: **The relationship between industrial technological progress and corporate risk-taking is important and interesting question. This paper makes a profound study through empirical research, as well as the mechanism of the role of digital financial development on the relationship between them. Through model and data analysis, it is concluded that industrial technological progress not only improves firms' ability to raise the level of risk-taking, but also generates three incentive effects through the incentive role of digital financial development, further expanding its impact on corporate risk-taking. According to regional and industry differences, the development of digital finance has a more pronounced positive moderating effect on industrial technological progress and risk-taking level among enterprises in the central and western regions as well as enterprises in high-tech industries.

Reviewer #2: No

---

## [Author Response · Author response to Decision Letter 0]

24 Jan 2024

Thanks to your suggestions, this article has been revised and the changes are marked in red.

---

## [Decision Letter · Decision Letter 1]

30 Jan 2024

Industrial Technology Progress, Digital Finance DevelopmentAnd Corporate Risk-Taking: Evidence from China's listed firms

PONE-D-23-41757R1

Dear Dr. Wu,

We’re pleased to inform you that your manuscript has been judged scientifically suitable for publication and will be formally accepted for publication once it meets all outstanding technical requirements.

Kind regards,

Han Lin

Academic Editor

PLOS ONE

Additional Editor Comments (optional):

Reviewers' comments:

Reviewer's Responses to Questions

**Comments to the Author**

1. If the authors have adequately addressed your comments raised in a previous round of review and you feel that this manuscript is now acceptable for publication, you may indicate that here to bypass the “Comments to the Author” section, enter your conflict of interest statement in the “Confidential to Editor” section, and submit your "Accept" recommendation.

Reviewer #1: All comments have been addressed

Reviewer #2: All comments have been addressed

2. Is the manuscript technically sound, and do the data support the conclusions?

Reviewer #1: Yes

Reviewer #2: Yes

3. Has the statistical analysis been performed appropriately and rigorously? 

Reviewer #1: Yes

Reviewer #2: Yes

4. Have the authors made all data underlying the findings in their manuscript fully available?

Reviewer #1: Yes

Reviewer #2: Yes

5. Is the manuscript presented in an intelligible fashion and written in standard English?

Reviewer #1: Yes

Reviewer #2: Yes

6. Review Comments to the Author

Reviewer #1: The revised manuscript has improved the information required for the submitted paper and has been revised according to the suggestions.

1. Related literatures were added to for measuring industrial technological progress by total factor productivity;

2. Detailed explanations of variables were given;

3. All the suggestions have been revised.

Reviewer #2: (No Response)

7. PLOS authors have the option to publish the peer review history of their article (what does this mean?). If published, this will include your full peer review and any attached files.

Reviewer #1: **Yes: **Chen Weimin

Reviewer #2: No

---

## [Editor Report · Acceptance letter]

7 Mar 2024

PONE-D-23-41757R1 

PLOS ONE

Dear Dr. Wu, 

I'm pleased to inform you that your manuscript has been deemed suitable for publication in PLOS ONE. Congratulations! Your manuscript is now being handed over to our production team.

Kind regards, 

on behalf of

Dr. Han Lin 

Academic Editor

PLOS ONE